# Measurements of heterogeneity in proteomics analysis of the nanoparticle protein corona across core facilities

Ali Akbar Ashkarran[1,9], Hassan Gharibi [2,9], Elizabeth Voke[3], Markita P. Landry [3,4,5,6], Amir Ata Saei [2,7,8] ✉ & Morteza Mahmoudi [1] ✉

Robust characterization of the protein corona—the layer of proteins that spontaneously forms on the surface of nanoparticles immersed in biological fluids—is vital for prediction of the safety, biodistribution, and diagnostic/therapeutic efficacy of nanomedicines. Protein corona identity and abundance characterization is entirely dependent on liquid chromatography coupled to mass spectroscopy (LC-MS/MS), though the variability of this technique for the purpose of protein corona characterization remains poorly understood. Here we investigate the variability of LC-MS/MS workflows in analysis of identical aliquots of protein coronas by sending them to different proteomics core-facilities and analyzing the retrieved datasets. While the shared data between the cores correlate well, there is considerable heterogeneity in the data retrieved from different cores. Specifically, out of 4022 identified unique proteins, only 73 (1.8%) are shared across the core facilities providing semi-quantitative analysis. These findings suggest that protein corona datasets cannot be easily compared across independent studies and more broadly compromise the interpretation of protein corona research, with implications in biomarker discovery as well as the safety and efficacy of our nanoscale biotechnologies.

Many technical factors contribute to poor reproducibility in nanomedicine, but the complexity inherent in the nanobio interface, of which the most heavily studied component is the protein corona, poses a major challenge[1]. To date, liquid chromatography coupled to mass spectroscopy (LC-MS/MS) remains the dominant methodology to characterize the protein corona in terms of the number of unique proteins and their relative abundances. However, the role of mass spectroscopy in causing technical variations in assessing the composition of the protein corona, and to what extent such proteomics outcomes can affect

the interpretation of the nanobio interfaces, remains largely unknown.

It is now well understood that the surface of nanoparticles (NPs), due to high surface free energy, becomes covered by a complex layer of biomolecules and mainly proteins, upon exposure to biological fluids[2,3]. This coating, known as the protein corona[3] (or eco-corona for environmental investigations)[4], defines the biological, chemical, and physical identities of NPs. Robust and accurate characterization of the protein corona is therefore essential in predicting interactions between NPs and biosystems, which define the safety and efficacy

[1]Department of Radiology and Precision Health Program, Michigan State University, East Lansing, MI, USA. [2]Division of Physiological Chemistry I, Department of Medical Biochemistry and Biophysics, Karolinska Institutet, Stockholm, Sweden. [3]Department of Chemical and Biomolecular Engineering, University of California, Berkeley, Berkeley, CA, USA. [4]Innovative Genomics Institute, Berkeley, CA, USA. [5]California Institute for Quantitative Biosciences, University of California, Berkeley, Berkeley, CA, USA. [6]Chan Zuckerberg Biohub, San Francisco, CA, USA. [7]Department of Cell Biology, Harvard Medical School, Boston, MA, USA. [8]Present address: Biozentrum, University of Basel, 4056 Basel, Switzerland. [9]These authors contributed equally: Ali Akbar Ashkarran, Hassan Gharibi. ✉e-mail: amirata.saei.dibavar@ki.se; mahmou22@msu.edu

outcomes of the biotechnologies built from those NPs[5–8]. Robust protein corona characterization plays a vital role in (i) the safe and efficient development and in vivo translation of nanobiotechnologies, (ii) the improvement of our fundamental understanding of the role of the protein corona in determining the biological fate of nanobiotechnologies, (iii) biomarker discovery and overcoming current critical issues in mass spectroscopy-based proteomics analysis of human plasma, and (iv) enabling comparison of independent protein corona studies.

The protein corona plays critical roles in the cellular uptake of NPs, recognition of functional moieties (e.g., antibodies) on the surface of NPs, and their interactions with the immune system[9–11]. The protein corona undergoes specific interactions during cell association followed by receptor-mediated membrane adhesion and subsequent cellular uptake[12]. It was recently reported that changing the NP soft corona to hard corona−thereby enabling turning of the dynamic nature of protein corona dissociation−showed a role in cellular uptake. It was found that certain "caretaker" proteins that comprise the hard corona mediate NP-cell interactions, which allows for higher-affinity interactions between the cell and NPs, resulting in different internalization routes[13]. Moreover, protein corona-aided targeting offers unique opportunities for specific drug delivery by manipulation of interaction modes of plasma proteins on the surface of NPs[14]. For instance, liposomes functionalized with short nontoxic amyloid beta (Aβ) peptides, can specifically interact with the lipid-binding domain of exchangeable apolipoproteins, consequently exposing the receptor-binding domain of apolipoproteins to achieve brain-targeted delivery[15]. In addition, the dynamic NP protein corona composition regulates the interaction of NPs with the physiological environment. It is shown that the formation of protein corona on the surface of liposomes considerably reduces their capture by circulating leukocytes in whole blood and may be an effective strategy to enable prolonged circulation in vivo[10].

Many biological and methodological factors can affect NP protein corona information in terms of protein type and abundance[5–8]. Despite efforts to minimize variability in protein corona studies, the effect of the "workhorse" technique LC-MS/MS on characterizing the protein corona remains understudied. This limited understanding of the role of LC-MS/MS in the variability of proteomics datasets characterizing the protein corona may lead to misinterpretation of the safety, diagnostic, and therapeutic efficacy of nanobiotechnologies and nanomedicines[7,8].

Generally, mass spectrometry-based proteomics provides robust and reproducible data, and any major differences between various experiments performed in different labs would be related to proteome coverage, i.e., the number of proteins quantified in a given sample or a set of samples. While such variations in the depth of analysis can introduce bias in data interpretation (for example, when not even detecting a low abundant genuine target), the bias can be minimized by increasing the proteome coverage. However, plasma proteomics is much more challenging than the analysis of most other biological materials, and the depth of analysis can vary significantly based on the workflow used[16]. The broad dynamic range of protein abundance in plasma is perhaps the main analytical difficulty[17]. Seven proteins constitute 85% of the total protein mass in plasma, with albumin alone making up 55%[18]. We also know that 22 proteins comprise 99% of plasma proteins by weight[19]. Peptides from such abundant proteins crowd the mass spectra and hamper the comprehensive coverage and in-depth analysis of plasma proteomes, especially for proteins with lower abundance. For this reason, several plasma depletion strategies have been introduced that exploit immunodepletion spin columns, immunodepletion-LC, magnetic beads, and even NPs themselves to deplete abundant or moderately abundant proteins from plasma[16,20,21]. However, such methods are unsuitable for use in large cohorts due to high cost, increased handling requirements (causing lower

reproducibility and underlying lower throughput), as well as carry-over concerns when processing multiple samples[22]. Another complication with the depletion of albumin is that many proteins are bound to albumin and can be co-depleted[23]. Apart from the dynamic range, another analytical difficulty is the presence of both known and unknown protein isoforms in plasma. Indeed, one of the emerging applications of NP protein corona is to reduce plasma proteome complexity in biomarker discovery[24,25].

In spite of these challenges, according to the human plasma proteome draft published in 2017, a compilation of data from 178 individual experiments shows that 3509 proteins have been reliably quantified in plasma, with a protein-level false discovery rate (FDR) of 1%[26]. Different studies have identified thousands of proteins with state-of-the-art MS-based plasma proteomics[27,28]. Advances in plasma proteomics are turning mass spectrometry into the tool of choice for biomarker discovery in large cohorts[29]. For example, in a recent study, 596 plasma samples were analyzed within 3 weeks to identify non-invasive proteomic biomarkers for alcohol-related liver disease[30]. In a recent paper, Ignjatovic et al.[16] provide considerations and recommendations concerning study design, plasma collection, quality metrics, plasma processing workflows, MS data acquisition, data processing, and bioinformatic analysis.

Protein corona analysis and plasma proteomics face similar challenges, as similar workflows are used for LC-MS/MS analysis. Since plasma proteomics is a field under continuous development, there are still no unified or streamlined protocols and workflows[16]. Furthermore, inevitably, different labs have access to different instrumentation, equipment, and software, which can also affect data processing workflows and relative protein quantification. The biological interpretation of data, especially in protein corona analysis, is highly dependent on the proteins detected or quantified; therefore, variabilities in the sheer number of quantified proteins and variations in data quality can impede the biological interpretation of data. Large variations in the analysis output in plasma proteomics can stem from the choice of sample preparation protocols, analytical columns, LC and MS systems, as well as the method settings and duration of the analysis. Small variations can also result from the choice of platform for the search and the sequence databases, and subsequent data processing. It would be highly desirable to identify the most important parameters contributing to the bias introduced by LC-MS in protein corona analysis, and to assess their effect on data interpretation.

In this work, to investigate the contribution of LC-MS/MS to the variations in the analysis and interpretation of protein corona data, we submit 17 identical aliquots of fully characterized protein corona samples to 17 proteomics core facilities in the United States and provide a comprehensive comparative analysis of the retrieved data. The results highlight the critical effect of LC-MS/MS workflow details (e.g., sample preparation protocol, instrumentation, and raw data processing) on protein corona results, which can create bias in interpretation of protein corona applications, including biomarker discovery and assessing the biological fate of nanomedicine technologies.

## Results

### Comprehensive characterization of NPs and protein coronas

Prior to studying the composition of the protein corona on the surface of the NPs using LC-MS/MS, all prepared samples (i.e., $n = 17$) were characterized using dynamic light scattering (DLS), zeta potential measurements, and transmission electron microscopy (TEM) to investigate the homogeneity/heterogeneity of the protein corona-coated NPs in each vial and to ensure the NPs had similar physicochemical properties (Fig. 1 presents the overall workflow of the study).

Supplementary Fig. 1 shows the size distribution histograms and the corresponding replicates of bare and protein corona-coated NPs used. The bare polystyrene NPs (PSNPs) have an average size of 78.8 nm and a surface charge of $-31.6 \pm 0.3$ mV (Supplementary Table 1

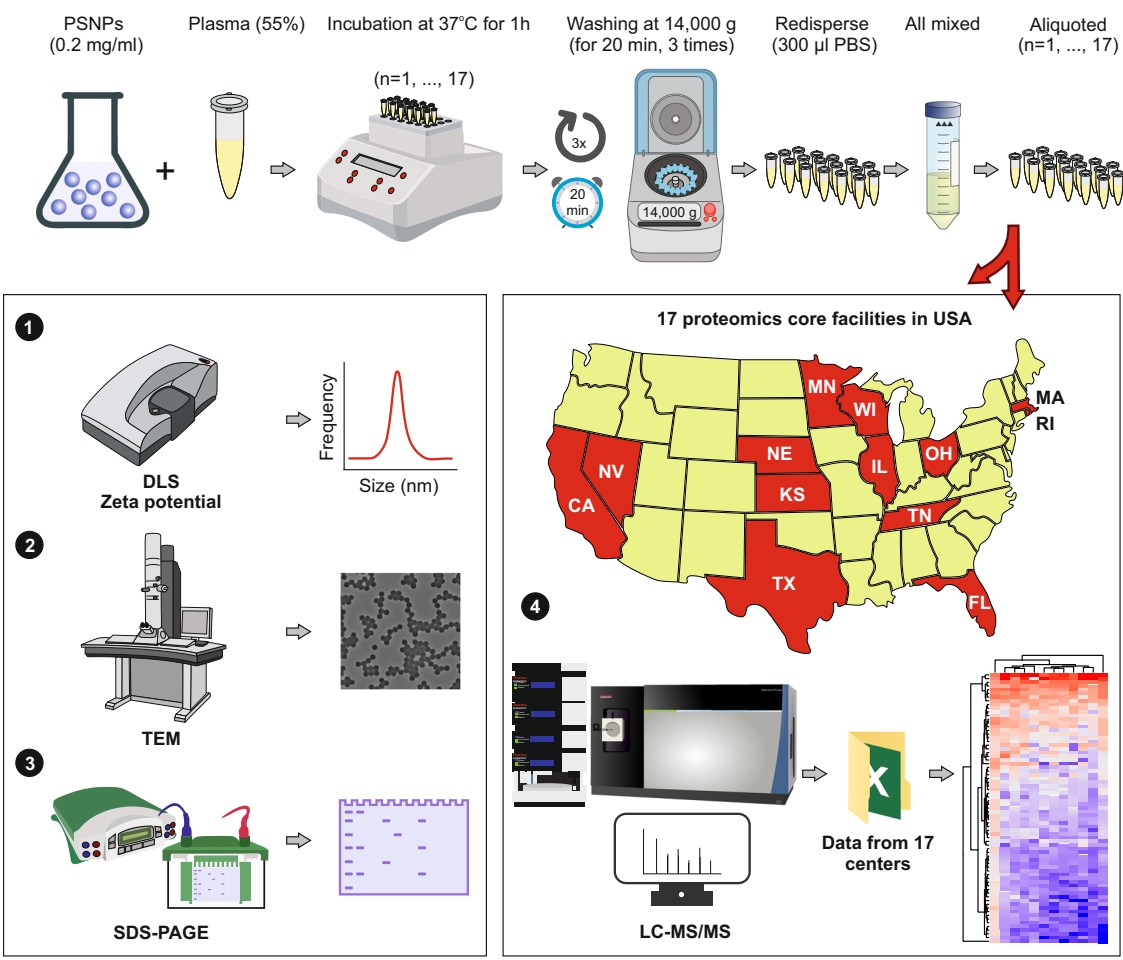

**Fig. 1 | Schematic representation of the study.** After formation of the protein corona-coated polystyrene NPs (PSNPs), 17 similarly prepared batches were characterized individually and shipped to various proteomics core facilities across the USA to investigate the homogeneity/heterogeneity of the proteomics data retrieved on the protein corona profile on the surface of NPs. PSNPs polystyrene nanoparticles, PBS phosphate-buffered saline, DLS dynamic light scattering, TEM transmission electron microscopy, SDS-PAGE sodium dodecyl-sulfate poly-acrylamide gel electrophoresis, LC-MS/MS liquid chromatography–mass spectrometry.

and Supplementary Fig. 1). As shown in Supplementary Table 1, the average size increased in all 17 samples with the formation of the protein corona on the surface of NPs. Our analysis revealed no significant difference in the average size values of the 17 aliquoted protein corona-coated NPs ($P$ value = 0.91; Supplementary Table 2). In addition to the hydrodynamic size of the NPs, as expected, the average surface charge of the NPs changed after the formation of the protein corona (Supplementary Table 1), and there were no significant differences in mean surface charge of the prepared samples ($P$ value = 0.65, Supplementary Tables 2 and 3).

TEM analysis was also used to assess the size of the NPs before and after the formation of the protein corona. The bare PSNPs used in this study were monodispersed and had a narrow size distribution confirmed by DLS and TEM analyses (Fig. 2a–c and Supplementary Figs. 1 and 2). As we and other groups have already shown, after exposure to a biological fluid (e.g., blood plasma), a thin layer of proteins forms on the surface of NPs (Fig. 2d–f)[31–34]. Furthermore, following the preparation of the 17 aliquots, 20 µl of the solution from each aliquot were removed, stained with uranyl acetate 1%, washed with deionized water, and characterized by TEM analysis. Supplementary Fig. 2 shows TEM images of protein corona-coated PSNPs of all 17 prepared samples. The stained protein corona-coated NPs demonstrate darker densities around the surface of the NPs (Fig. 2d). As expected, all imaged aliquots revealed the presence of a darker shell on the surface of the NPs, indicating formation of the protein corona. Such a dynamic

biomolecular shell significantly alters the primary physicochemical properties of the NPs and consequently their biological identities[35].

In addition to traditional TEM, cryo-TEM images of the protein corona-coated NPs revealed formation of pure and fully coated individual NPs (without any trace of aggregation) with proteins from human plasma (Fig. 2e, f). As cryofixation preserves the original state of the hydrated biological and colloidal dispersions, the features observed in the cryo-TEM images in Fig. 2f likely represent the initial state of the NPs and their associated protein corona. The configuration of the NPs, and their corresponding protein corona preserved in the vitreous ice layer of the holey carbon TEM grids, reflects the true nature of the protein corona and their adsorption onto the NPs.

Qualitatively, no differences in protein corona formation were observed in TEM-imaged samples.

## Proteomics analysis of protein corona aliquots

We next submitted the 17 identical aliquots to 17 core facilities across the USA for analysis (see Supplementary Information for the links to each core facility): Harvard University, Stanford University, Massachusetts Institute of Technology (MIT), Case Western Reserve University, Wayne University, University of Illinois, Cornell University, University of Tennessee, University of Nebraska-Lincoln (UNL), University of Missouri, University of Cincinnati, University of Florida, University of Kansas Medical Core (KUMC), University of Texas at San Antonio (UTSA), Michigan State University (MSU), University of

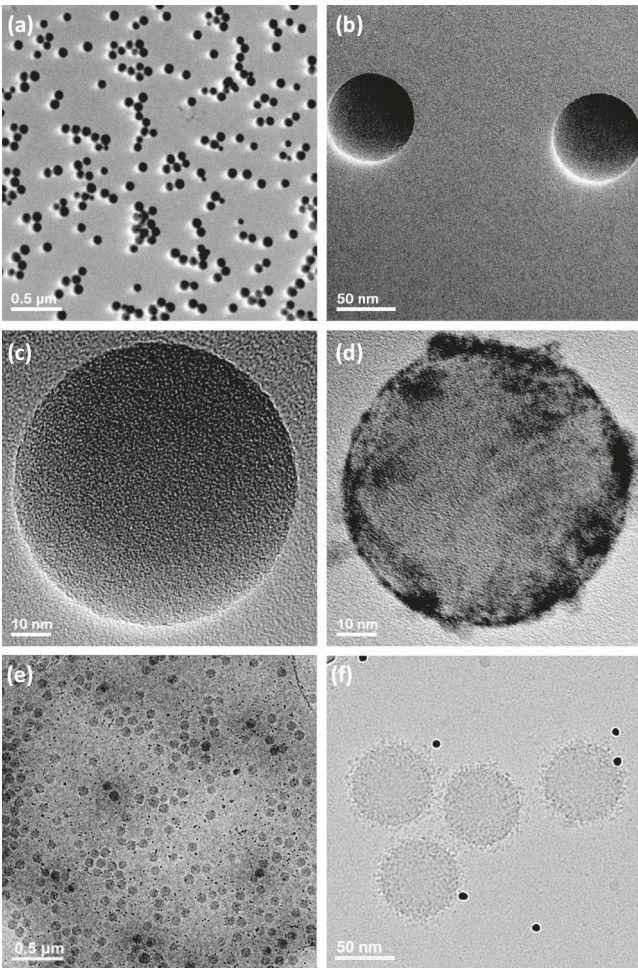

**Fig. 2 | Electron microscopy characterization.** TEM images of (**a**–**c**) bare PSNPs, (**d**) protein corona-coated PSNPs that clearly reveal proteins on their surface after incubation with human plasma proteins, (**e**) and (**f**) Cryo-TEM images of protein corona-coated PSNPs obtained with a direct electron detector and phase plate, clearly showing the distribution of proteins on the surface of the PSNPs (black dots in the images (**e**, **f**) are 10 nm gold fiducial markers). The results are representative of three independent measurements.

California San Diego (UCSD) and University of Nevada, Reno (UNR). We requested analysis of the protein corona with 3 technical replicates. The usual process for nanomedicine labs is to submit their corona-coated samples to mass spectrometry core facilities and request proteomics data using the discretion and/or commonly used protocols of those particular core facilities. We requested highly detailed protocols and reports from each core facility, including protein and peptide level intensities. Hereafter, in this report, we blind the core facility names with random numbers to avoid any possible conflict of interest. Of these, 12 core facilities provided semiquantitative protein intensities in their reports, and 5 provided data based on peptide spectrum match or total spectra count. We therefore opted to analyze the semi-quantitative data from 12 core facilities in-depth and provide a concise report on the other 5 cores.

Out of a combined dataset of 4022 unique proteins identified across all 12 cores (Supplementary Data 1), only 73 unique protein IDs were shared, representing a 1.8% protein ID similarity across identical samples. We next performed a full comparative analysis of the proteomics data from the 12 cores. To avoid bias in our comparative analysis of detected proteomes, we did not remove any data points such as contaminants, alternative splice isoforms, variable region immunoglobulin sequences, or missing values, as there was no

standardized approach to remove them uniformly from datasets of different cores. The protein counts, peptide counts, coefficients of variation (CV) of technical replicates, as well as the median sequence coverages are compared in Fig. 3a. Peptide counts can be essential when analyzing post-translational modifications (PTMs) and were thus included in our analysis. Reproducibility of measurements and low CVs are especially important in supporting actionable clinical decisions based on proteomics data. Our results revealed large variations between protein corona analyses of the various proteomics core facilities. Various facilities performed differently with regards to the number of quantified peptides/proteins, and CV or sequence coverage. For some cores, variations in CV were large, given that the analysis was based on technical replicates. Overall, cores 2, 3, 5, 9, and 11 provided higher-quality data with regards to the four parameters mentioned above. Figure 3b shows that by including more cores in the merged dataset, the number of missing values incrementally increases, as expected. The normalized intensities of 73 shared proteins are shown as a heatmap in Fig. 3c. Except for core 1, the other cores (11) show good reproducibility for these 73 proteins. The overall distribution of normalized protein and peptide abundances is shown for comparison in Fig. 3d, e, respectively.

To better show the correlation of the obtained data across different facilities in an unbiased manner, in Fig. 4, the correlation of data for the 73 shared proteins among the cores is shown. Overall, with regards to the shared proteins, all cores reported highly comparable data. The data from most of the cores show a correlation >0.7 with those of the other cores. Only cores 1 and 6 show correlations <0.7 with some other cores, recapitulating the demonstrated results in the heatmap presented in Fig. 3c. These acceptable correlation levels among different cores demonstrate that the variability mainly originates from varying coverage obtained due to variations in protocols, workflows and raw data processing. Therefore, obtaining higher proteome coverage is essential for more accurate interpretation of protein corona data and the elimination of associated bias. We performed a similar analysis for the 5 cores that performed better than others with regards to the four criteria. Out of 1778 unique proteins detected cumulatively, 151 (8.5%) proteins were shared across the 5 cores, which is an improvement compared to the analysis of 12 centers. The data from these cores also showed generally better correlation with each other as compared to those of the 12 cores (Supplementary Fig. 4).

The upset plot shown in Fig. 5 shows the uniqueness of the data obtained from the 12 different core facilities. This plot was made with all the proteins quantified by all the cores and highlights the fact that the differences (uniqueness) of the data from each core outweigh the similarities between the data from different cores.

We also plotted dynamic range (defined as the ratio of most intense protein divided by least intense protein in data from each core) vs. protein count to determine whether there were any correlations between the two parameters (Supplementary Fig. 5). The higher protein counts were to some extent associated with higher dynamic range ($R^2$ of 0.33).

## Variations in experimental workflow among core facilities

The detailed protocols used by each core are included in the Supplementary Information file. Overall, there were many differences in sample preparation, instrumentation, quantification methods, search parameters, and raw data processing and reporting among different cores. While some of the protocols are highly detailed, some are very concise and omit important parameters.

In Supplementary Table 4, we summarize the main similarities and differences that can create variability in the analysis output. In the current study, using a high-end mass spectrometer and duration of the LC method were not always associated with a better coverage of

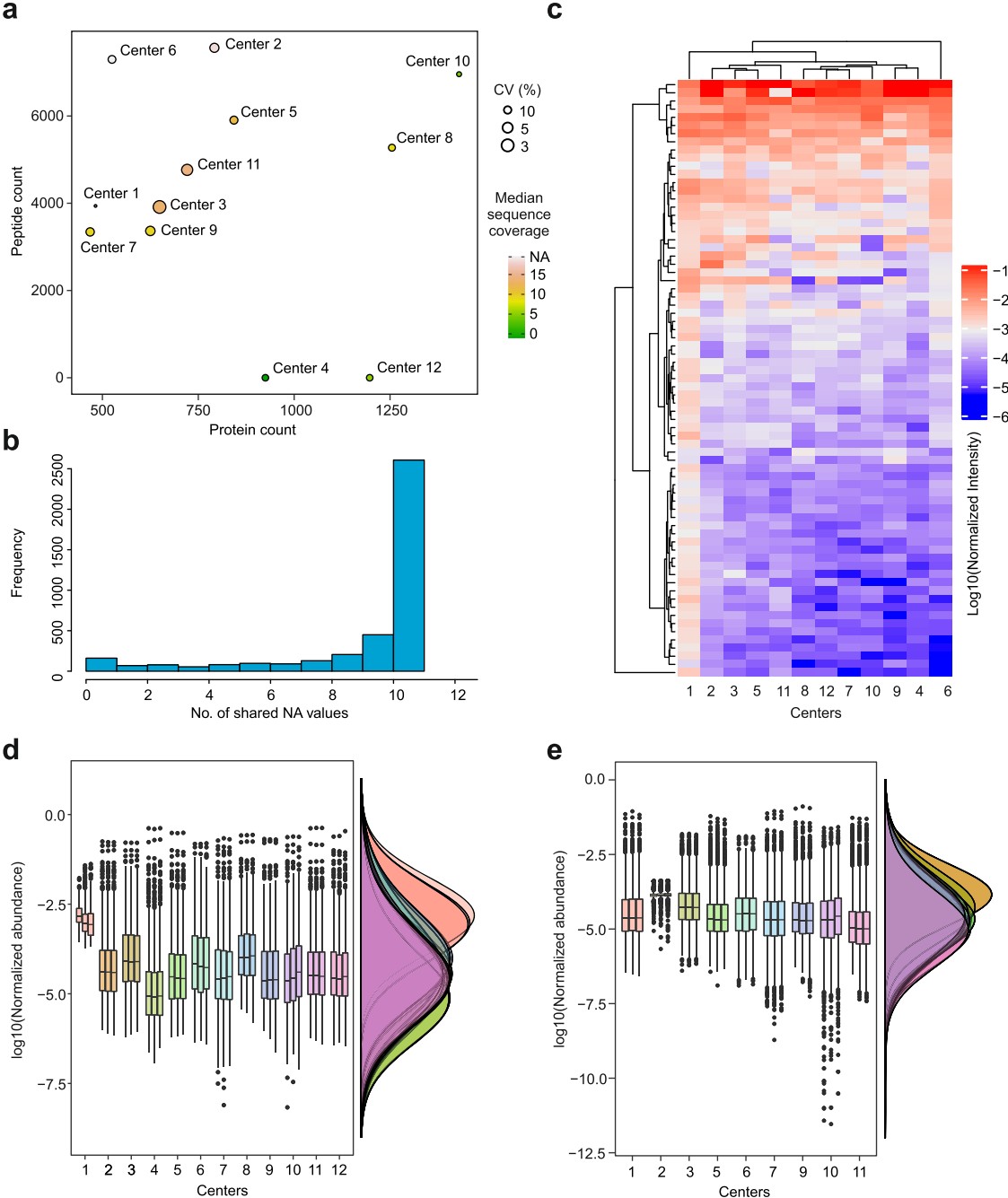

**Fig. 3 | Comparative protein corona proteomics analysis by the 12 cores that provided semiquantitative protein intensity information. a** Comparing the average peptide count (peptide count was not provided by cores 4 or 12), average protein count, median coefficients of variation CV (%) between the technical replicates and median protein sequence coverage (%) for 12 core facilities. **b** The number of missing values expectedly increased when data from different cores were merged. **c** Heatmap of 73 shared proteins with any number of peptides between the 12 cores. **d** Distribution of protein-level intensities for the 12 cores. **e** Distribution of peptide level intensities for 10 cores, as cores 4 and 12 did not provide peptide level intensities (center line, median; box limits contain 50%; upper and lower quartiles, 75 and 25%; maximum, greatest value excluding outliers; minimum, least value excluding outliers; outliers, more than 1.5 times of upper and lower quartiles). All analyses were based on three technical replicates.

the proteome, high median sequence coverage, or lower CVs, indicating that other parameters such as sample preparation protocols as well as methodological and instrumental settings also play crucial roles in shaping the data output. The effect of different parameters on the output between different centers can be directly compared in Fig. 4 and Supplementary Table 4. In the majority of proteomics workflows, methionine oxidation and acetylation of protein N-termini are included as variable modifications. Another source of variation in data processing were the 4 cores that did not even report the modifications

included in the database search. On the other hand, some cores included more modifications for the database search; for example, core 5 included phosphorylation of serine, threonine, and tyrosine. Moreover, while the routine procedure is to include up to 2 missed cleavages with trypsin digestion, cores 5 and 17 allowed for 3 and 5 missed cleavages, respectively. Finally, cores 6 and 8 performed semi-specific searches. There were also large differences in how different cores handled the FDR at the peptide and protein levels. Though such changes in search parameters do not undermine the validity of any

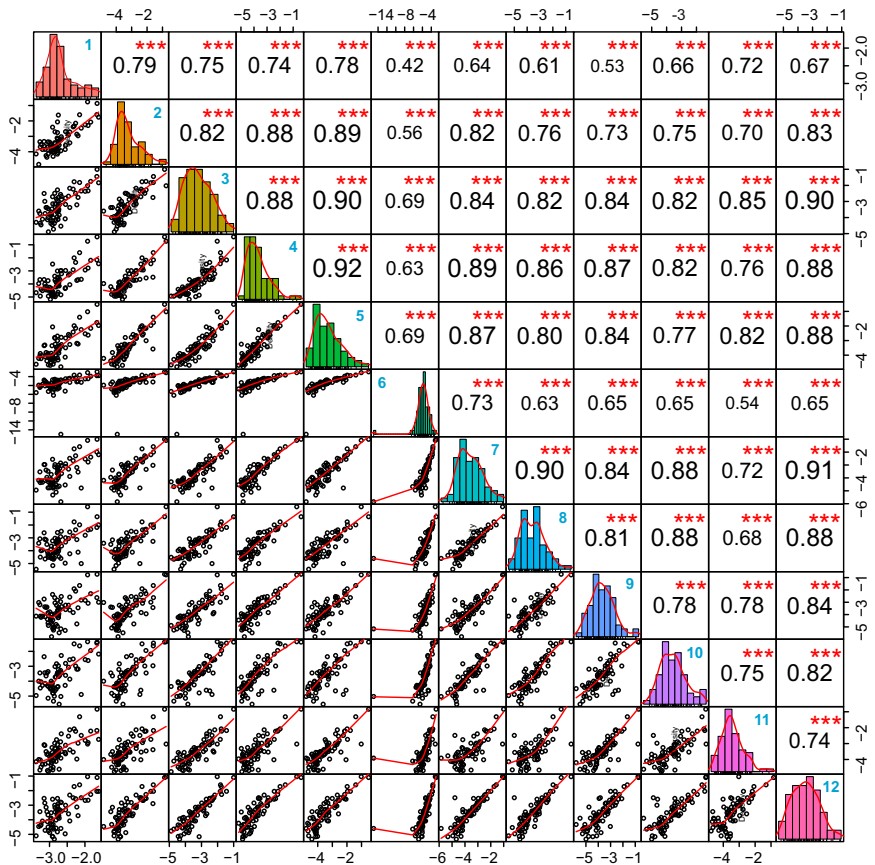

**Fig. 4 | Pearson correlations of protein intensities across 12 cores based on 73 shared proteins.** Mean normalized intensities of three technical replicates for each core were used for correlation analysis. The correlation coefficient is given in black text and core codes are given in bold blue text in the top right corner of the boxes. *** denote a significance below *P* value of 0.001 (derived from Pearson correlation analysis).

given study here, we highlight these variations in parameters as a source of heterogeneity.

Finally, for the 5 cores not providing semiquantitative data, the summary is also provided in Supplementary Table 4 and compiled data is provided in Supplementary Data 2. Center 17 is not included in this Supplementary Data 2, as the report was provided at the peptide level. Furthermore, center 13 only reported a single value for each protein, and we have included these data in Supplementary Data 2 as is.

## Discussion

The protein corona forms spontaneously on NPs intended for use in biological tissues and fluids. Given the known role of the protein corona in affecting the outcome of bionanotechnologies and nanoscale medicines, much effort has been focused on characterizing the protein corona in terms of protein identity and abundance. Despite the numerous reports available in the literature concerning the protein corona, efforts to reconcile independent studies and consolidate protein corona datasets toward predicting protein corona composition and NP biocompatibility outcomes remain limited[36]. Most efforts thus far have focused or methodological differences in protein corona formation conditions or protein isolation and recovery protocols. There are even variabilities in measurements of simple parameters such as NP size. For example, a study investigating NP size among 12 different QualityNano laboratories using blinded samples showed a higher level of variation and lack of reproducibility when no protocol was supplied and when no training of participants was performed[37]. In the mentioned study, providing a well-established protocol and standard operating procedures (SOPs) helped reduce CV of measurements

among the laboratories involved. However, it should be noted that measurement of a single parameter such as NP size involves far fewer steps than the steps involved in a LC-MS/MS workflow. It would be very difficult, if not impossible, to perform comparative proteomics analysis in a similarly controlled manner, as not all labs have access to the same instruments, equipment, and commercial software.

Despite the central role of LC-MS/MS in providing proteomics datasets for NP protein corona studies, the variabilities generated by LC-MS/MS itself remain understudied. Biological and technical variations can significantly compromise the reproducibility of proteomics analyses. The community is becoming more aware of the parameters influencing the detectability of proteins, affecting data outcomes[6,38–40]. For example, a recent paper demonstrated that biological variations in different stocks of cultured human HeLa cells can lead to huge heterogeneity in proteomics analysis and even partially determine the phenotypic response of different cell lines to *Salmonella* infection[41].

A few studies have also investigated technical variations in the analysis of proteome samples among laboratories, albeit not involving NP-based protein corona formation. In one such study, the reproducibility of Sequential Window Acquisition of all THeoretical fragment ion spectra (SWATH) MS data acquisition was evaluated among 11 sites worldwide[42]. The sample under study was a set of standard peptides with serial dilutions spiked into HEK293 cell extract. In this study, all sites used the same SCIEX TripleTOF 5600/5600+ mass spectrometer systems, while the nanoLCs consisted of various models from the same vendor (SCIEX), but sometimes different chromatographic columns were used, although they had the same dimensions (30 cm × 75 µm). In the aggregated data analysis, where the database search and FDR

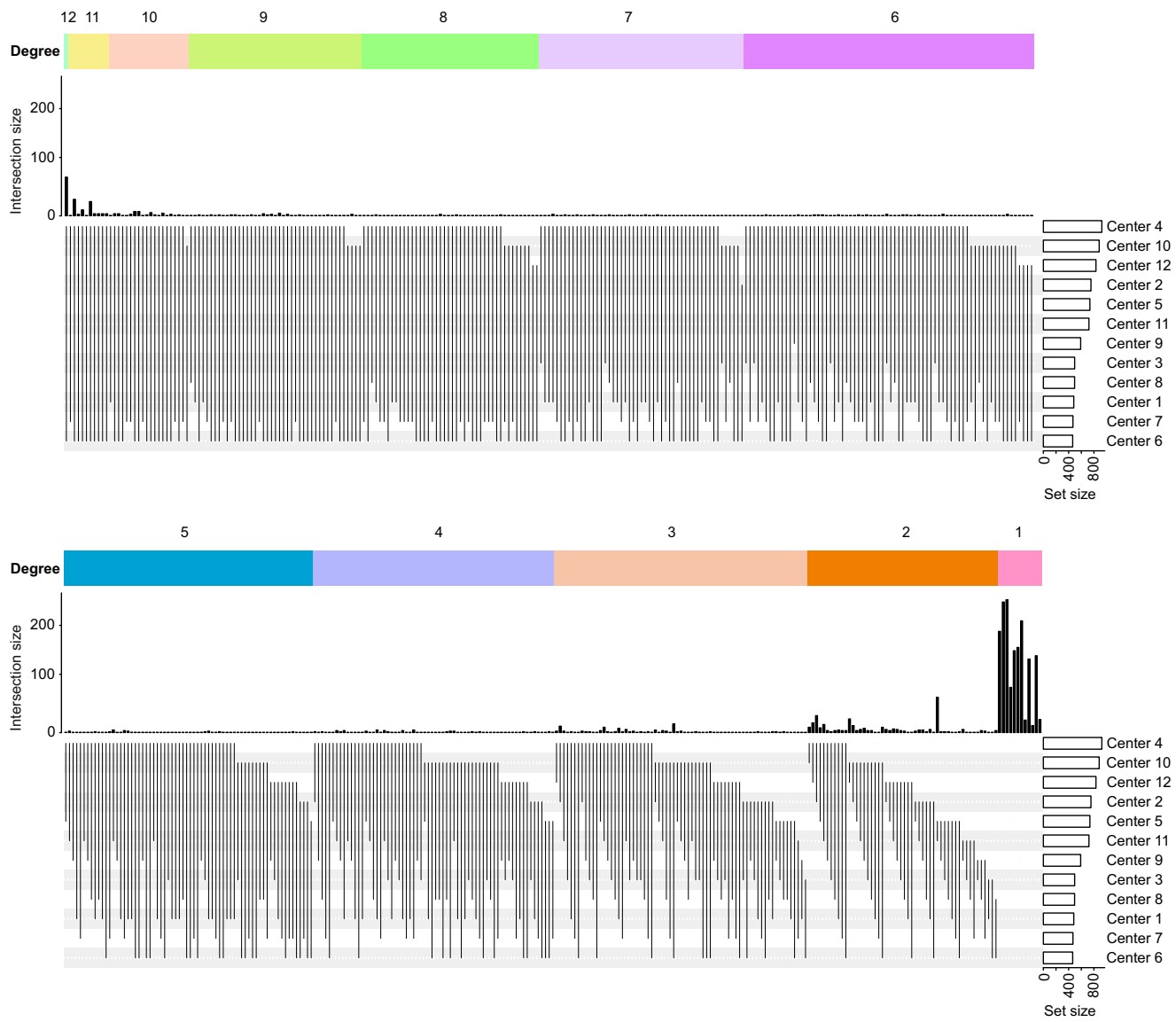

**Fig. 5 | Uniqueness of the data obtained from the 12 different core facilities.** Upset plot showing the large variabilities in the number of detected proteins within the data provided by 12 cores.

control was performed in a centralized way, a core set of 4077 proteins were consistently detected in >80% of all samples analyzed across 11 sites. A fraction of proteins was not detected in all sites. Furthermore, when the analysis and FDR control were carried out independently on a site-by-site basis, there was a reduction in consistently detected proteins among 11 sites. One important consideration is that there was no sample preparation involved in the above study and therefore, large batch effects were not expected. This study shows that if sample preparation is centrally performed, and instrumental parameters are kept largely constant, it is possible to achieve reproducible and comparable quantification of the majority of proteins in a complex proteomics dataset.

In our study, all experimental steps involving NP-protein corona formation were performed identically, such that samples supplied to mass spectrometry centers were identical. However, once received, the protein corona samples were prepared for subsequent mass spectrometry analysis on-site, no standardized instrumentation parameters or protocols were requested/provided, and the data extraction and analysis were not aggregated. Furthermore, as stated earlier, plasma-based samples are generally more challenging to analyze than cell or tissue samples using MS. Last but not least, the cores in our

study had to employ data-dependent acquisition (DDA). Generally, more variability is expected in the data-dependent acquisition, since unlike the data-independent acquisition method used in the above study (SWATH), data-dependent acquisition involves stochastic fragment ion (MS2) sampling[43]. In DDA, when the number of precursor selection cycles are exceeded by the number of precursor ions[44], selection of precursors occurs stochastically, leading to lower repeatability of peptide sampling[43,45].

Another study between two centers has shown that given a standardized affinity purification–mass spectrometry (AP-MS) workflow for 32 kinases, high interlaboratory reproducibility (81%) can be achieved (albeit upon centralized data analysis and several data filtering steps), in spite of differences in MS configurations and subtle variations in sample preparation protocols[46]. Interestingly, it was found that sample preparation was 12% less reproducible than MS between the two laboratories. It is noteworthy that the study was carried out over a period of 3 years, during which time the authors studied and defined experimental parameters, exchanged protocols, chose the appropriate reagents, and used consumables with the same lot numbers. Furthermore, the scientists involved were trained to perform the workflows. Similar interlaboratory studies on other

technologies such as RNAseq and microarrays have also shown that the analysis platforms and data analysis pipelines can result in large differences in data output[47,48].

Due to the resemblance of protein corona samples to plasma, lessons can be learned from studies on plasma proteomics. Various studies have assessed the effects of preanalytical sample processing and storage on the integrity of plasma proteome for analysis, such as the effects of biobanking conditions[49–51], delays in plasma preparation[38,52,53], multiple freeze-thawing cycles[38,53], postcentrifugation delay in time and temperature[50,51,54], hemolysis[38,55] and carry-over of platelets and coagulation[55], centrifugation force[50–52], and differences in proteomics of EDTA-plasma vs. heparin-plasma[56]. Furthermore, in 2016, the Association of Biomolecular Resource Facilities (ABRF) Proteome Informatics Research Group (iPRG) investigated variability in proteoform inference and FDR estimation from bottom-up proteomics data among different research labs[57].

Focusing on the technical variations that may be introduced by sample preparation protocols and proteomics analysis, here we present substantial data heterogeneity in protein corona analysis of an identical sample performed by 17 different core facilities and discuss how bias may be introduced in the biological interpretation of protein corona studies (e.g., on biomarker discovery and/or studies on the biological fate of NPs). Data provided by different cores (we discuss semiquantitative data from only 12 cores) differ with regard to the number of quantified proteins (ranging from 467 to 1430), number of quantified peptides (ranging from 1541 to 7565), median CV (ranging from ~3 to ~32% between technical replicates), and median protein sequence coverage (ranging from 4 to 19%). Across 12 cores, only 1.8% of the proteins were shared in identical samples. Such a high variation across LC-MS/MS centers compromises interpretation of independent NP protein corona studies, and even calls into question inter-study dataset comparisons, especially if those datasets were not analyzed by the same LC-MS/MS facility. For instance, there are numerous studies claiming differences in NP protein corona compositions as a function of NP size, shape, charge, surface chemistry, etc., yet these sample comparisons may hold true only if LC-MS/MS dataset integrity can be confirmed. Furthermore, interpretation of protein corona datasets as they pertain to the biological fate of NPs may be questionable, particularly for analyses based on comparisons of independent studies. It should however be stated that the proteins shared among the centers usually have a higher abundance than other detected proteins in the corona. It is possible that more abundant proteins play the main roles in masking functional moieties and in affecting NP uptake, transport, or their interactions with cells, which benefits from the relative consistency across centers in quantifying the more abundant proteins in the NP corona. However, recent studies have also shown that NPs often show preferential protein adsorption or depletion in their coronas, such that even high-abundance plasma proteins such as albumin are so strongly depleted in the NP corona they can just barely be measured with mass spectrometry[58]. Results such as the latter highlight the importance of consistency across proteomics centers not just for high-abundance proteins but for lower-abundance proteins as well. Another consideration is that, since tubes or other materials used for sample processing can enhance or deplete the presence of specific proteins (perhaps especially proteins with low abundance), negative or mock samples without NPs can be included in the workflow and analyzed in parallel to rule out proteins that do not necessarily belong to protein corona, or that conversely have been artificially depleted due to sample handling protocols.

There is an acceptable correlation between the proteins quantified across 12 cores, showing that proteomics methodology is not the main factor behind variability in the analysis results; instead, it is different sample preparation routines, instrumental settings, raw data processing and a range of other confounding factors that significantly affect the proteome coverage and impose a bias. For example, as mentioned above, in DDA, there are also inevitable variabilities introduced by stochastic MS2 sampling[43]. Some of these variabilities can arise from different strategies used in protein recovery from NPs, as previous research has shown that using on-particle, in-solution and in-gel digestion of the protein corona can affect both the class of proteins recovered and their abundances measured in protein corona[59]. While during on-bead digestion, protease cleavage sites may not be accessible due to the orientation of proteins at the NP surface, during in-gel digestion, proteins and peptides can be lost through the isolation process[59]. As such, these mentioned technical variabilities produce different depths of analysis among the cores, which obviously affects the biological interpretation of any given study. This is especially important in protein corona research, as each protein could serve as a biomarker of safety, diagnosis, or prognosis. Therefore, using better protocols and optimizing instrumental parameters would ensure higher coverage of the proteome, better representing the underlying biology and bringing interpretations closer to reality. It is noteworthy that the proteomics of other biological samples such as cell and tissue extracts is much less biased, as high coverage of the proteome can be reached due to lower protein dynamic range and less challenging nature of such samples. One could argue that we could request a deeper analysis of the protein corona samples, but since nanomedicine labs submit their corona samples requesting proteomics data using the discretion and/or commonly used protocols of the given core facility, we opted to do the same. Also, deeper analysis would usually require labeling, multiplexing and prefractionation of samples before LC-MS/MS analysis, introducing more steps in sample preparation and further complicating the comparative analysis of the results. We also acknowledge that the results obtained by different core facilities would have been more homogenous if we requested the core facilities to follow the same methodology using similar instrumentation and protocols. However, such a study would not recapitulate the heterogeneity of NP protein corona datasets currently in existence.

While these data should not be generalized to plasma proteomics, our results partly recapitulate those obtained by the 2017 human plasma proteome draft[26]. Reanalyzing the data collected from 178 individual experiments from 2005 till 2017, only 50% of the studies reported the 500 most abundant plasma proteins[26].

We acknowledge that comparing across institutions is challenging. This is because data analysis is not standardized yet, and most institutions are able to report only relative abundance rather than semiquantitative abundance. The latter type of data can be more easily compared. Nevertheless, in our study, the quality of analyses performed by cores 2, 3, 5, 9, and 11 was higher than others with regards to the number of detected proteins and peptides as well as CV of technical replicates and protein sequence coverage. While we have blinded the core names in the current paper, the associated protocols can be found in Supplementary Information, which can be adapted and exploited for more comprehensive and reproducible protein corona proteomics.

Human blood and blood-derived products (such as plasma and NP protein corona) are viable proxies for the identification of biomarkers and translation of research results into the clinic. With continuously emerging applications of protein corona analysis in the assessment of NPs' safety and predicting their biological fate as well as biomarker discovery for disease detection, based on these data, as a community we need to pursue paths to minimize and report these differences so that we can better understand the underlying bias in proteomics data for protein corona experiments. This study can, therefore, serve as the first step in unifying and streamlining a standardized mass spectroscopy approach involving sample preparation protocols and instrumental settings for protein corona analysis[60], similar to guidelines suggested for plasma proteomics[57].

In the meantime, we recommend precise documentation of protocols when reporting protein corona data. We would like to raise

awareness of the challenges to be addressed and hope that the current study will fuel community discussions on how experimental data from protein corona proteomics should be analyzed, reported, and interpreted. Such efforts will further enhance the validity of studies in protein corona proteomics, which, in turn, could significantly improve the clinical translation of both diagnostic and therapeutic nanomedicine products.

The results of each study can be generally trusted, as long as all the samples (for example, those of healthy and patient groups) are analyzed in the same core. Our findings do not undermine the results obtained by any core facility, but rather highlight that the analysis of a given sample by different cores can result in different data interpretations. For example, with low depth of analysis, some important biomarkers might be missed, or biomarkers with less importance come into focus instead. Furthermore, minimizing technical noise is critical in identifying biologically significant changes, especially those of small size.

In summary, our pilot study demonstrates that the characterization of protein corona formed on NP surfaces is subject to high variability originating from the analysis and compilation of MS data. Broadening the comparison to similar protein descriptions could provide a better understanding of the differences between proteomics core facilities, while relying on MS data being provided in a format amenable to descriptive and statistical comparison. Nonetheless, these results underscore the variability of protein corona data originating from proteomics core facilities alone and emphasize the fact that this degree of variability is larger than the effect size of most published NP/protein corona literature. Therefore, particularly when protein corona data are compared using analysis from multiple proteomics core facilities, the interpretation of effect size is more likely to originate from core-to-core variability rather than biological effect size. Regardless, experimental and analytical validation should be an integral part of protein corona studies. Our study will hopefully pave the way for developing best practice and quality control measures in NP protein corona research by developing gold standard protocols and workflows in LC-MS/MS sample preparation and analysis.

# Methods

## Materials

Healthy human plasma protein was obtained from Innovative Research (www.innov-research.com) and diluted to a final concentration of 55% using phosphate buffer solution (PBS, 1X). Plain PSNPs (~‘100 nm) were provided by Polysciences. (www.polysciences.com).

## Protein corona formation on the surface of NPs

For protein corona formation, NPs were incubated with 55% plasma (with NP concentration of 200 μg/ml) for 1 h at 37 °C at a constant agitation (total volume: 17×1.5 mL Eppendorf tubes). To remove unbound and plasma proteins only loosely attached to the surface of NPs, protein-NP complexes were then centrifuged at 14,000×$g$ for 20 min, the collected NPs' pellets were washed twice more with cold PBS under the same conditions, and the final pellet was redispersed at 300 μl of PBS in each tube. All fully washed protein corona-coated NPs were then mixed, shaken, and aliquoted again to 17 Eppendorf tubes for further analysis. All prepared protein corona-coated NP samples were shipped overnight using FEDEX small standard 4 °C units (to avoid extra freezing and thawing)[61] with guaranteed next-morning delivery. It is noteworthy that the units had 48 h of cooling time at 4 °C and safe arrival of all samples was confirmed by each core facility.

## Characterization

DLS and zeta potential analyses were performed to measure the size distribution and surface charge of the NPs before and after protein corona formation using a Zetasizer nano series DLS instrument (Malvern company). A Helium Neon laser with a wavelength of 632 nm was used for size distribution measurement at room temperature. Protein corona profiles at the surface of the NPs were studied by sodium dodecyl sulphate–polyacrylamide gel electrophoresis (SDS-PAGE).

TEM was carried out using a JEM-2200FS (JEOL Ltd.) operated at 200 kV. The instrument was equipped with an in-column energy filter and an Oxford X-ray energy dispersive spectroscopy (EDS) system. In total, 20 μl of the bare PSNPs was deposited onto a copper grid and used for imaging. For protein corona-coated NPs, 20 μl of the sample was negatively stained using 20 μl uranyl acetate 1%, washed with DI water, deposited onto a copper grid, and used for imaging at the same day. Details of the cryo-TEM protocols are available in ref. 62. Protein corona composition was also determined using LC-MS/MS. LC-MS/MS analyses were caried out at 17 different proteomics cores across the USA. It is noteworthy that DLS, zeta potential, SDS-PAGE, and LC-MS/MS analyses were performed for protein corona-coated NPs taken from the same vial.

## LC-MS/MS sample preparation and proteomics data processing

Details of sample preparation, LC-MS/MS analysis, and data processing protocols are included in the Supplementary Information file for each core and summarized in Supplementary Table 4. To avoid bias, we requested standard plasma analysis from each core in three technical replicates, and cores were allowed to follow their own protocols and workflows. From each core, we requested a highly detailed protocol describing each step of the analysis. The protocols in the Supplementary Information file have been only slightly revised for consistency (e.g., "hour" was changed to "h").

## Data analysis

First, for each core, data were normalized by total protein intensity in each technical replicate. CVs were calculated based on normalized intensities between technical replicates for each protein. To unify protein IDs, for some cores, the various protein IDs used were converted to UniProt IDs. The data among the cores were combined by UniProt IDs. Data analysis was performed using R project version 3.6.1.

## Statistics and reproducibility

All centers performed a triplicate analysis of the given aliquot.

## Reporting summary

Further information on research design is available in the Nature Research Reporting Summary linked to this article.

# Data availability

Due to the blinding of core names in the current study, and since the MS .raw files can be traced, the raw data and associated individual data files are available upon request from corresponding authors (A.A.S. and M.M.). The extracted protein abundance data and relevant outputs of data analysis are provided in the supplementary data files cited in the text. Supplementary Data 1 was used to generate Figs. 3–5 and Supplementary Figs. 4 and 5 (Supplementary Data 1 is the Source Data for all proteomics analyses). Different core facilities used various software, including Scaffold v.4.11.1, v.5.1, and v5.0.1, Proteome Discoverer 2.4, 2.4.0.305 and P2.2.0.388, PEAKS-XPro server, Peaks Studio 8.5, PEAKS Studio 10plus, Byonic v4.2.4, Sequest, MSFragger, Mascot 2.8, and Mascot Distiller v2.7.0 for data extraction. Different core facilities used the following databases including SwissProt TaxID 9606 downloaded on v.2017/05/10 with 42,153 entries, Swissprot database downloaded on 2021/02, UniProt human database (UP000005640) downloaded on 12/11/2020, UniProt downloaded on 07/02/2019, UniProt-Human database updated on 03/08/2021 with 20,379 entries, UniProt-human_20210508 database with 77,027 entries, UniProt (UP000005640) downloaded on 03/30/2021 with 20,310 protein

entries, UniProt-homo_sapiens_20190201.fasta with 147,857 entries, NCBI, and NcbiAV TaxID=9606 downloaded on v2017/10/30. Details are given in Supplementary Table 4.

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

## Acknowledgements

This work is supported by the U.S. National Institute of Diabetes and Digestive and Kidney Diseases (DK131417). The University of Cincinnati Proteomics Laboratory acknowledges NIH high-end instrumentation grant (S10OD026717-01). A.A.S. was supported by the Swedish Research Council (grant 2020-00687) and the Swedish Society of Medicine (grant SLS-961262, 1086 Stiftelsen Albert Nilssons forskningsfond). We acknowledge all 17 core facilities for their efforts and contributions to the current study.

## Author contributions

Conceptualization, A.A.A., A.A.S., and M.M.; project organization, resources, and funding acquisition, A.A.S. and M.M.; methodology and experiment design, A.A.A., A.A.S. M.P.L., and M.M.; experiments and characterizations, A.A.A.; data analysis and visualization, H.G., A.A.S., E.V., and A.A.A.; writing—original draft, A.A.A., A.A.S., H.G., M.P.L., and M.M.; writing, review & editing, all co-authors.

## Competing interests

Morteza Mahmoudi discloses that (i) he is a co-founder and director of the Academic Parity Movement (www.paritymovement.org), a non-profit organization dedicated to addressing academic discrimination, violence, and incivility; (ii) he is a Founding Partner at Partners in Global Wound Care (PGWC); and (iii) he receives royalties/honoraria for his published books, plenary lectures, and licensed patents. The remaining authors declare no competing interests.
