## [Peer Review File · Nature Communications]

Measurements of heterogeneity in proteomics analysis of nanoparticle protein corona across core facilitiesREVIEWER COMMENTS

Reviewer #1 (Remarks to the Author):

In the manuscript "Measurements of heterogeneity in proteomics analysis of nanoparticle protein corona across core facilities", Ali Akbar Ashkarran and colleagues describe the analysis of the nanoparticles plasma protein corona across 17 proteomics core facilities. The same aliquots of nanoparticles were distributed across the laboratories and the investigation lead to identification of 4022 unique proteins. The laboratories involved applied different instrumentation and SOP and only the 1.8% of proteins were identified in all the laboratories.

I think the paper is very interesting and in a sense educational, since it shows how sophisticated technologies as mass spectrometry despite being very powerful should be used with greater awareness.

However, I believe that authors conclusions are only partially appropriated, therefore this study cannot be published as it is and I suggest the authors to perform a major revision of their manuscript re-considering the data from a different perspective. The study can be interpreted as a comprehensive characterization of the protein corona using complementary methods, while it may be difficult to come any conclusions regarding reproducibility and heterogeneity since the study was obviously not designed to evaluate reproducibility.

An interesting point highlighted here is the importance of method consistency to get reproducible results, which is however a concept well known in clinical chemistry, and this is why analytical methods that wish to be used in clinical or pre-clinical settings need appropriate validation (including repeatability and reproducibility studies).

Major concerns:

1. The authors seem to have the expectation that the data across the different laboratories should be highly reproducible, however this is not a reproducibility study since the authors agreed with the facility on using different protocols, different instruments and methods for chromatography. Mass spectrometry based methods can be highly reproducible and robust, but as any analytical methods changing major variables such as instrument, chromatography, etc can dramatically change the results. This is something that the authors could predict.
2. Despite the samples distributed across the labs were prepared with the same protocol is not clear at which temperature they were stored after preparation. See paragraph "Protein corona formation on the surface of NPs".
3. Here storage and shipment temperature may have influenced sample variability. In the material and method authors state "Protein corona composition was also determined using liquid chromatography mass spectroscopy (LC-MS/MS). LC-MS/MS analyses were carried out at 17 different proteomics cores across the US. For LC-MS/MS analysis the prepared protein corona-coated NPs were shipped overnight at 4 °C (using FEDX small standard 4 °C units with 48 hours cooling time) with guaranteed next-morning delivery." 4C it maybe a high temperature to store and ship plasma proteins (even if in the corona) and this detail could have caused some variability of the actual protein corona in first instance. Usually plasma samples are stored at -20/-80C. the author should discuss their choice.

4. The author discuss literature about plasma proteomics however, considering the topic of this study, they should include more literature about reproducibility of MS analysis intra and inter-laboratory and discuss the design of their study in that perspective.

Some examples here:

Varjosalo M et al. Interlaboratory reproducibility of large-scale human protein-complex analysis by standardized AP-MS. *Nat Methods*. 2013 Apr;10(4):307-14. doi: 10.1038/nmeth.2400. Epub 2013 Mar 3. PMID: 23455922

Ben C Collins et al. Multi-laboratory assessment of reproducibility, qualitative and quantitative performance of SWATH-mass spectrometry. *Nat Commun*. 2017 Aug 21;8(1):291. doi: 10.1038/s41467-017-00249-5.

5. In the discussion the authors state: "One could argue that we could request a deeper analysis of the protein corona samples, but since nanomedicine labs submit their corona samples requesting proteomics data using the discretion and/or commonly used

protocols of the given core facility, we opted to do the same.” I think that one mostly could argue that the design of this study was not clear. If the authors wanted a reproducibility study then they should have requested the laboratories to apply the same instruments, method for digestion, peptide purification, chromatography and mass spectrometry analysis. When mass spectrometry is applied in medicine lab, for the sake of reproducibility, the same exact SOP is applied. What the author could do here is to select a set of relevant proteins that could be relevant to analyze in a clinical setting to predicting interactions between NPs and biosystems, and then develop for those quantitative mass spectrometry methods (with internal standards) and test the reproducibility of quantification of those proteins across lab using the same analytical methods.

6. Table S4. “Comparison of the protocols, LC and MS systems, method duration, and other important parameters between all the cores.”, shows that some of the lab had similar instrumentation. Did the authors investigate similarity between the results produced from the lab using the same instrumentation at least? Or same length of chromatography?

7. Regarding the variability in data search, this problem has been overcome, by re-analyzing the raw data from different experiment under the same search platform (eg. Protein Discoverer). Could the authors get the raw file and re-analyze them applying uniform data search parameters?

8. “Various facilities performed differently with regards to the number of quantified peptides/proteins, and CV or sequence coverage. For some cores, CV variations were large, given that the analysis was based on technical replicates. Overall, cores 2, 3, 5, 9, and 11 provided higher-quality data regarding to the four parameters mentioned above.” How and for how long the samples were stored upon received by the facilities 2, 3, 5, 9, and 11?

Minor comment

1. What do the authors mean with higher-end mass spectrometer? Maybe, high-end?

2. Author stated “Of these, 12 core facilities provided quantitative protein intensities in their reports, and 5 provided data based on peptide spectrum match or total spectra count. We therefore opted to analyze the quantitative data from 12 core facilities in depth and provide a concise report on the other 5 cores.” Please correct across the text the use of the word “quantitative”, mass spectrometry is quantitative only when internal standards are applied, which seems not to be the case here. In any other case is a semi-quantitative method.

3. In the discussion, the meaning of the following sentence is not clear : “There is acceptable correlation between the proteins quantified across 12 cores, showing that proteomics methodology is not the main factor behind variability in the analysis results; instead, it is different sample preparation routines, instrumental settings, and raw data processing that significantly affect outcomes.”

4. Please revise the first part of the results text. It seems that the same concept is repeated several time (such as “It is noteworthy that DLS, zeta potential, and TEM characterizations were performed to investigate the homogeneity/heterogeneity of the protein corona-coated NPs in each vial (Figure 1))”.

Reviewer #2 (Remarks to the Author):

The manuscript entitled „Measurements of heterogeneity in proteomics analysis of nanoparticle protein corona across core facilities“ is a benchmark comparison of protein corona samples which were measured in parallel by 17 proteomic core facilities. The manuscript is very well written and give the experimental conditions and the results in great detail (or at least as detailed as provided by the different core facilities obviously). I therefore highly support the publication of the manuscript in this form. I have the following comments/suggestions:

1. The authors’ view on the protein corona is mostly for identification of biomolecular markers for diseases. Although the protein corona has been used for this in previous studies there is/are (an)other field(s) for the protein corona which I think should be mentioned in the manuscript. Actually there are more publications about the protein

corona and the consequences of the protein coating for masking the chemically defined surface – also targeting moieties like antibodies, ligands etc. –, the uptake into cells and the cellular reactions towards proteins of the protein corona. The authors should point that out more clearly in my view and not narrow the field of protein corona only to biomarker studies.

2. As a consequence: LC-MS in the different core facilities found several thousand protein in total (4022), but “only” 73 (1,8%) are shared by at least by most of the core facilities. The authors describe this to be disappointing or at least not as optimal as it could be. The authors attribute this low number correctly in my view to preprocessing of the samples before the LC-MS proteomics and also to the circumstances how the primary data is analyzed by the bioinformatics tools. I would stress here that 1. At least for cellular uptake and cellular effects it is most likely that the most abundant proteins will play a role while protein number #433 on the list (sorted descending according to abundance in the protein corona) probably does not contribute at all. In fact, all relevant proteins in the field of masking, cellular uptake, cellular function etc. is found in the list of 20-50 most abundant proteins in the list (most of the times it will be even one of 10 most abundant proteins). I think that the authors should point this out.

3. As a side note: I even think that protein #47 to #4022 represent in many cases proteins which were enhanced by the plastic tube or the Eppendorf tip or other materials or process parameters in the workup of the individual core facility. Therefore, the authors may recommend that “negative” or mock samples should always be prepared in parallel, i.e. samples where no nanoparticles are present and therefore only material and conditions of the process are taken into account (when we do this we found also up to several hundred irrelevant proteins – their hallmark is their low abundance compared to the real protein corona proteins).

Reviewer #3 (Remarks to the Author):

It is important to support people and teams that are trying to do a good job for a community, and take on a challenge that few have mastered. In my opinion we should therefore think carefully about the value of these things before lightly discouraging them. The paper presented here makes some excellent points, and should be encouraged.

The whole question of reproducibility was discussed some years ago, in areas from antibody validation, to biomarkers, and even protein coronas. The area is just so complex, I will summarise briefly.

I have some comments the authors might like to consider.

Firstly, many of these types of blind round robins were attempted earlier on in teams from across US and EU, and much was learned. Some of the conclusions were not very palatable, and as the work was sometimes not regarded as 'research' the reports were not always published in the most accessible places. In the nano area one of the most simple measurements and known round robins is Journal of Nanoparticle Research volume 15, Article number: 2101 (2013), and it is instructive to see how that was prepared, developed and done. Similar things were done for coronas, and many other biological outcomes such as toxicity.

The outcomes were much as reported here, and a lot of efforts was put into understanding just why. Sadly, it transpired that even the most accomplished laboratories were often well off the standard answer in early phases, and this lead to a process of protocol and training between the engaged groups. This eventually lead to convergence between all of the labs for nearly every quantity-the particle size example illustrates, including high overlap for the protein corona lists of hits, and even rankings. This established that such outcomes were possible, but it was found that even the arduous efforts with protocols, in the absence of training between labs produced poor

outcomes. It should be noted that the particle size example is one of the most simple, and many of the painters there invented the instruments, or developed them and were certainly outstanding!

This leads to the uncomfortable that many measurements in science are poorly reproduced by teams of scientists, no matter how solid the lab, and that something extra is needed. This is a difficult message, and I think one hard for anyone to send, and likely it is implicit in what the authors are saying here. The most experienced labs in certain areas control an enormous number of factors within the labs, and others do not. Some labs get good results in certain biomarker assays, some do not until there is a process of protocol evolution, handover and training. All of this was also discussed at length within the Nature Reproducibility Initiative, and largely speaking all was clear.

The question is how to pass the message in this particular example, and for this generation of people. Other report and many other scientists have shown how controls can be imposed (<https://hal.archives-ouvertes.fr/hal-02163525/document>, <https://www.ncbi.nlm.nih.gov/pmc/articles/PMC6631359/>). While I would not agree with all of the content of these many of these reports and discussions, the overall outcomes are sort of common and agreed. Certainly I think the authors do not wish to say that it is impossible to get reproducible answers between labs (we know this is not the case) nor that they as individuals are not able to succeed. After all, these are highly talented and excellent people, that could do so if they pushed ahead with the kind of rouble robin processes outlined above.

I am sure they have in mind that many results reported results are unreliable, and are illustrating this very well. Quite frankly, some labs are able to get good biomarker reproducibility in all fields and others are not, and the reasons are many. This is a very hard message to pass, as it can appear very critical. In my experience the truth of this issue is extremely difficult to communicate.

It will take maybe a little though to see how to refine the message of the paper to give some lasting value. I would allow the authors to refine their message a bit. But certainly they are making highly valuable points. And the fact the points have been made before and not fully absorbed may mean it is time for a good journal to take on the task to help?

REVIEWER COMMENTS

Reviewer #1:

In the manuscript “Measurements of heterogeneity in proteomics analysis of nanoparticle protein corona across core facilities”, Ali Akbar Ashkarran and colleagues describe the analysis of the nanoparticles plasma protein corona across 17 proteomics core facilities. The same aliquots of nanoparticles were distributed across the laboratories and the investigation lead to identification of 4022 unique proteins. The laboratories involved applied different instrumentation and SOP and only the 1.8% of proteins were identified in all the laboratories. I think the paper is very interesting and in a sense educational, since it shows how sophisticated technologies as mass spectrometry despite being very powerful should be used with greater awareness.

However, I believe that authors conclusions are only partially appropriated, therefore this study cannot be published as it is and I suggest the authors to perform a major revision of their manuscript re-considering the data from a different perspective.

The study can be interpreted as a comprehensive characterization of the protein corona using complementary methods, while it may be difficult to come any conclusions regarding reproducibility and heterogeneity since the study was obviously not designed to evaluate reproducibility.

An interesting point highlighted here is the importance of method consistency to get reproducible results, which is however a concept well known in clinical chemistry, and this is why analytical methods that wish to be used in clinical or pre-clinical settings need appropriate validation (including repeatability and reproducibility studies).

We thank the reviewer for the positive and constructive comments. We have addressed all comments point-by-point and the corresponding changes are shown in blue color in the revised manuscript.

We agree with the reviewer that our study was (intentionally) not designed to measure reproducibility in the classic sense (variability within the same core facility), but rather to measure heterogeneity in proteomics analysis performed across cores. It is becoming increasingly prevalent in the NP protein corona literature to compare studies or build research off of prior work, and our intent is to highlight the degree to which inter-core dataset variability can undermine such efforts. Specifically, the aim of the study was to estimate the variation in proteomics outcomes of identical nanoparticle protein corona samples among different proteomics centers. It would also be very challenging if not impossible to do such reproducibility studies among 17 centers, since most labs would not have access to the similar instrumentation (mass spectrometers and chromatography systems would need to be purchased/upgraded for certain centers, for example), similar software (such as commercial database search software), neither they could/would comply with following exact protocols (for example using different enzymes for digestion, reducing agents, cleaning protocols, etc). Additionally, it would also be challenging to measure the protocol compliance level, e.g., incubation times, quality of materials used, etc. As discussed above, the motivation of

our work is to highlight the “real world” situation in which samples are provided to a core facility for analysis, and data is recovered by the user for interpretation – usually with minimal to no additional intervention. However, the reviewer makes a good point that the way we have framed our results misses this underlying motivation. Therefore, we have attenuated our discussions on the reproducibility related statements throughout the manuscript. We also acknowledge that reproducibility can have a broad definition; for example reproducibility of a whole study where completely different techniques might be used to reach the same conclusions. We have now tried to clarify our message throughout the manuscript, in light of the reviewer comments. We have also added reviewer’s insight on the importance of validating results acquired by mass spectrometry-based proteomics.

Major concerns:

1. The authors seem to have the expectation that the data across the different laboratories should be highly reproducible, however this is not a reproducibility study since the authors agreed with the facility on using different protocols, different instruments and methods for chromatography. Mass spectrometry based methods can be highly reproducible and robust, but as any analytical methods changing major variables such as instrument, chromatography, etc can dramatically change the results. This is something that the authors could predict.

We agree that mass spectrometry-based methods can be highly reproducible and robust, and we have discussed this issue at length in the manuscript, to avoid undermining the method. We did not expect very high reproducibility even when submitting identical nanoparticle protein corona samples to different proteomics centers due to some obvious reasons such as variations in instrumentation, different protocols and materials used for isolation and extraction of the proteins, etc. However, even we were surprised to find only 1.8% protein identity similarities for our otherwise identical protein corona samples submitted. Despite the evident differences in protocols and instrumentations across centers (that will undoubtedly contribute to low reproducibility), this surprisingly low amount of protein identity overlap highlights the issue at hand and does make it difficult if not impossible to compare proteomics datasets across NP protein corona research if the data were acquired at different core facilities.

Moreover, many protein corona laboratories that are not specialized in mass spectrometry-based proteomics rely on core facilities for proteomics analysis of their samples. Usually, such analysis is performed in the core facility of the same university/research organization to take advantage of internal costs and avoid the cost and time associated with training/onboarding and running their own samples. Many universities and core facility sites also prohibit or make difficult running one’s own samples due to the high cost of instrument maintenance and repair associated with mass spectrometry equipment. As a result, many researchers are not aware of the heterogeneities that might arise from proteomics analysis routines in different core facilities. As we have mentioned in the text, the results of all core facilities are correct, and it is just the interpretation of the results that can change. This interpretation bias is felt more in protein corona studies, as the mere presence and absence of proteins can significantly affect the interpretation of biological findings. As stated in the manuscript, the heterogeneity is much less in non-plasma related samples and the bias is reduced by quantitation of a higher number of proteins for example in cells and tissues. The bias

is never eliminated though, as a low abundant target of a drug can be simply missed due to low coverage of routine proteomics too. Again, our intent is not to say that mass spectrometry is not useful for NP protein corona studies (which would be untrue), but that comparing existing studies against each other or consolidating datasets on protein corona research when acquired from different core facilities may not represent scientific best practice. Thus, in this paper, we specifically focus on the heterogeneity of proteomics measurements of protein corona, and we have tried to fully incorporate the reviewer's insight in the revision.

2. Despite the samples distributed across the labs were prepared with the same protocol is not clear at which temperature they were stored after preparation. See paragraph. "Protein corona formation on the surface of NPs".

Immediately after preparation, samples were transferred to separate FEDEX small standard 4 °C units and submitted for overnight shipping to core facilities by FEDEX, and all arrived safely the next morning. These details are now added to the revised manuscript in the "Protein corona formation on the surface of NPs" section. It is noteworthy that safe arrival of all samples was confirmed by all core facilities in the morning.

3. Here storage and shipment temperature may have influenced sample variability. In the material and method authors state "Protein corona composition was also determined using liquid chromatography mass spectroscopy (LC-MS/MS). LC-MS/MS analyses were carried out at 17 different proteomics cores across the US. For LC-MS/MS analysis the prepared protein corona-coated NPs were shipped overnight at 4 °C (using FEDX small standard 4 °C units with 48 hours cooling time) with guaranteed next-morning delivery." 4C it maybe a high temperature to store and ship plasma proteins (even if in the corona) and this detail could have caused some variability of the actual protein corona in first instance. Usually plasma samples are stored at -20/-80C. the author should discuss their choice.

That is correct and storage temperature have a crucial effect on the composition of the protein corona, and we have also covered this issue in our manuscript. Long term storage of protein corona coated NPs is done at -20 °C but for short-term uses like the current study, 4 °C is preferred since it is reported that the protein corona composition remains intact at 4 °C at least for 48 hours (see reference 62) and freezing (storing again at -20 °C) and thawing the proteins may result in changes in protein corona composition of the NPs or damage to the proteins. Furthermore, freeze thawing is more likely to induce NP aggregation, which could affect the results. In addition, most proteomics centers either started processing the samples after arrival (depending on their schedules) or stored the samples at -20 °C upon arrival. We ensured safe sample arrival in the morning at 4 °C and then the core facility was responsible for taking care of the process/analysis (simulating the real scenario where the users do not have control over the remainder of the process).

4. The author discuss literature about plasma proteomics however, considering the topic of this

study, they should include more literature about reproducibility of MS analysis intra and inter-laboratory and discuss the design of their study in that perspective. Some examples here: Varjosalo M et al. Interlaboratory reproducibility of large-scale human protein-complex analysis by standardized AP-MS. *Nat Methods*. 2013 Apr;10(4):307-14. doi: 10.1038/nmeth.2400. Epub 2013 Mar 3. PMID: 23455922

Ben C Collins et al. Multi-laboratory assessment of reproducibility, qualitative and quantitative performance of SWATH-mass spectrometry. *Nat Commun*. 2017 Aug 21;8(1):291. doi: 10.1038/s41467-017-00249-5.

We thank the reviewer for highlighting these relevant papers. We have now fully discussed these papers in the text. We have also discussed and compared the design of our study in comparison to the similar papers in the discussion section.

5. In the discussion the authors state: “One could argue that we could request a deeper analysis of the protein corona samples, but since nanomedicine labs submit their corona samples requesting proteomics data using the discretion and/or commonly used protocols of the given core facility, we opted to do the same.” I think that one mostly could argue that the design of this study was not clear. If the authors wanted a reproducibility study then they should have requested the laboratories to apply the same instruments, method for digestion, peptide purification, chromatography and mass spectrometry analysis. When mass spectrometry is applied in medicine lab, for the sake of reproducibility, the same exact SOP is applied. What the author could do here is to select a set of relevant proteins that could be relevant to analyze in a clinical setting to predicting interactions between NPs and biosystems, and then develop for those quantitative mass spectrometry methods (with internal standards) and test the reproducibility of quantification of those proteins across lab using the same analytical methods.

We agree with the reviewers' comment and have now toned down our discussions on the reproducibility related statements throughout the manuscript. We added in the discussion “We also acknowledge that the results obtained by different core facilities would have been more homogenous, if we asked the core facilities to follow the same methodology using similar instrumentation and protocols. For example, a study conducted for investigating NP size among 12 different QualityNano laboratories showed a higher level of variation and lack of reproducibility when no protocol was supplied and no training of those involved was performed. In the mentioned study, providing a well-established protocol and standard operating procedures (SOPs) helped reduce CV of measurements among the laboratories involved. However, it should be noted that measurement of a simple parameter such as NP size involves far less steps than the steps involved in a LC-MS/MS workflow.”

It is noteworthy that the main aim of this study was neither evaluating the reproducibility of MS technique nor the protein corona itself and that is why a standard protocol was not delivered to various cores (representing a “typical” case of core analysis of a user sample). The main aim was to investigate how much variation in proteomics analysis of MS outcomes may occur when identical samples are submitted to various proteomics cores for mass spectroscopy analysis. It was

obvious that if we had asked core facilities to follow an identical protocol, we might have observed higher similarities in results, as reported previously (see for example references 42 and 46). Moreover, most researchers in the field of protein corona research prepare and submit the samples to proteomics centers for MS analysis without asking the cores to follow a specific procedure. Most of the proteomics centers have their own internal protocols and standards which includes a multi-step sample preparation using a wide range of materials at different experimental conditions. In addition, most of the available proteomics centers are established for years and have their own infrastructure, chemical inventories, instrumentation, experimental protocols and sample preparation routines. Therefore, any changes in any parts of the experimental procedure, sample preparation, materials, protocols, etc. may cause a significant variation by itself in the final results of the different cores, as the personnel might not have been trained for different steps of the workflow.

6. Table S4. “Comparison of the protocols, LC and MS systems, method duration, and other important parameters between all the cores.”, shows that some of the lab had similar instrumentation. Did the authors investigate similarity between the results produced from the lab using the same instrumentation at least? Or same length of chromatography?

We appreciate the reviewer’s comments- We do touch upon this in the same section stating that “In the current study, using a high-end mass spectrometer and duration of the LC method were not always associated with a better coverage of the proteome, high median sequence coverage, or lower CVs, indicating that other parameters such as sample preparation protocols as well as methodological and instrumental settings also play crucial roles in defining the data output”. We have added the number of proteins and peptides along with CV and sequence coverage in **Supplementary Table 4**, which can be used to compare the results for the same instruments and all other parameters. In addition, all such comparisons between the cores can be made in **Fig. 4**, where all possible correlations are depicted.

7. Regarding the variability in data search, this problem has been overcome, by re analyzing the raw data from different experiment under the same search platform (eg. Protein Discoverer). Could the authors get the raw file and re-analyze them applying uniform data search parameters?

We considered this possibility when preparing the manuscript, but to avoid bias in our interpretations, we decided not to reanalyze the data using the same platform (e.g., Proteome Discoverer). This would ultimately go against our aim to investigate the heterogeneity of various sample and data processing workflows. We also minimized the steps in our data analysis platform to reduce the bias in our report, as any manipulation of the results would add further bias in our results and report. One more bias is the choice of the particular search platform, as different cores have opted to work with five different software: Proteome Discoverer, PEAKS, Mascot, MSfragger and Byonic. There is no consensus on which database search platform is “the best”. The search parameters are also sometimes drastically different among the facilities, as some labs chose to add different sets of variable modifications such as phosphorylation, used up to 5 missed cleavages in their searches and some even searched for semi-tryptic peptides. We have also failed in retrieving the raw files from some of the cores. Collectively, we decided that this search might

be biased by our choice of “what is the best search platform and what are the right parameters?” and that would probably create more questions than answers.

8. “Various facilities performed differently with regards to the number of quantified peptides/proteins, and CV or sequence coverage. For some cores, CV variations were large, given that the analysis was based on technical replicates. Overall, cores 2, 3, 5, 9, and 11 provided higher-quality data regarding to the four parameters mentioned above.” How and for how long the samples were stored upon received by the facilities 2, 3, 5, 9, and 11?

Before sending the samples to the different cores, we already arranged with all cores to send the sample on a single day and date, using FEDEX overnight shipping at 4 °C, with guarantee next day morning delivery. We also asked all centers to process the sample immediately upon arrival and do the analysis (in total, we communicated with more than 30 cores to find those that would agree to our terms), but a few of them still were not able to do the analysis as agreed. While most of the cores processed the samples upon arrival, some could not perform the analysis right away, and therefore stored the sample at -20 °C.

Minor comment

1. What do the authors mean with higher-end mass spectrometer? Maybe, high-end?

The reviewer is correct. We have now corrected this term in the manuscript.

2. Author stated “Of these, 12 core facilities provided quantitative protein intensities in their reports, and 5 provided data based on peptide spectrum match or total spectra count. We therefore opted to analyze the quantitative data from 12 core facilities in depth and provide a concise report on the other 5 cores.” Please correct across the text the use of the word “quantitative”, mass spectrometry is quantitative only when internal standards are applied, which seems not to be the case here. In any other case is a semi-quantitative method.

We have now corrected the terminology throughout the manuscript.

3. In the discussion, the meaning of the following sentence is not clear : “There is acceptable correlation between the proteins quantified across 12 cores, showing that proteomics methodology is not the main factor behind variability in the analysis results; instead, it is different sample preparation routines, instrumental settings, and raw data processing that significantly affect outcomes.”

We have rephrased and clarified this sentence and it now reads: “These acceptable correlation levels among different cores demonstrate that the variability mainly originates from varying coverage obtained due to variations in protocols, workflows and raw data processing. Therefore, obtaining higher proteome coverage is essential for more accurate interpretation of protein corona data.”.

4. Please revise the first part of the results text. It seems that the same concept is repeated several time (such as “It is noteworthy that DLS, zeta potential, and TEM characterizations were performed to investigate the homogeneity/heterogeneity of the protein corona–coated NPs in each vial (Figure 1))”.

We have removed the redundancy and revised the text according to the reviewer comment.

Reviewer #2:

The manuscript entitled „Measurements of heterogeneity in proteomics analysis of nanoparticle protein corona across core facilities“ is a benchmark comparison of protein corona samples which were measured in parallel by 17 proteomic core facilities. The manuscript is very well written and give the experimental conditions and the results in great detail (or at least as detailed as provided by the different core facilities obviously). I therefore highly support the publication of the manuscript in this form. I have the following comments/suggestions:

We thank the reviewer for the positive comments.

1. The authors’ view on the protein corona is mostly for identification of biomolecular markers for diseases. Although the protein corona has been used for this in previous studies there is/are (an)other field(s) for the protein corona which I think should be mentioned in the manuscript. Actually there are more publications about the protein corona and the consequences of the protein coating for masking the chemically defined surface – also targeting moieties like antibodies, ligands etc. –, the uptake into cells and the cellular reactions towards proteins of the protein corona. The authors should point that out more clearly in my view and not narrow the field of protein corona only to biomarker studies.

We thank the reviewer for the comment. We have now added more discussion in the introduction of the revised manuscript regarding other aspects of the protein corona research such as effect of protein corona on cellular uptake of NPs, effects on targeting, etc.

2. As a consequence: LC-MS in the different core facilities found several thousand protein in total (4022), but “only” 73 (1,8%) are shared by at least by most of the core facilities. The authors describe this to be disappointing or at least not as optimal as it could be. The authors attribute this low number correctly in my view to preprocessing of the samples before the LC-MS proteomics and also to the circumstances how the primary data is analyzed by the bioinformatics tools. I would stress here that 1. At least for cellular uptake and cellular effects it is most likely that the most abundant proteins will play a role while protein number #433 on the list (sorted descending

according to abundance in the protein corona) probably does not contribute at all. In fact, all relevant proteins in the field of masking, cellular uptake, cellular function etc. is found in the list of 20-50 most abundant proteins in the list (most of the times it will be even one of 10 most abundant proteins). I think that the authors should point this out.

We agree with the reviewer. We have now added statements in the discussion section of the revised manuscript clarifying these issues. The paragraph now reads: “The proteins shared among the centers usually have a higher abundance than other detected proteins in the corona. Since most abundant proteins play the main roles in masking functional moieties and impeding NP uptake or their interactions with cells, such studies on NP uptake and interactions will be less affected (than biomarker studies) by heterogeneity of measurements observed between facilities, as highly abundant proteins are routinely detected.”.

3. As a side note: I even think that protein #47 to #4022 represent in many cases proteins which were enhanced by the plastic tube or the Eppendorf tip or other materials or process parameters in the workup of the individual core facility. Therefore, the authors may recommend that “negative” or mock samples should always be prepared in parallel, i.e. samples where no nanoparticles are present and therefore only material and conditions of the process are taken into account (when we do this we found also up to several hundred irrelevant proteins – their hallmark is their low abundance compared to the real protein corona proteins).

We agree with the reviewer and have added this statement to the discussion of the revised manuscript. It reads: “Another consideration is that, since the tubes or other materials in sample processing can enhance the presentation of specific proteins (perhaps especially low-abundance ones), negative or mock samples without NPs can be included in the workflow and analyzed in parallel to rule out proteins that do not necessarily belong to protein corona.”

Reviewer #3:

It is important to support people and teams that are trying to do a good job for a community, and take on a challenge that few have mastered. In my opinion we should therefore think carefully about the value of these things before lightly discouraging them. The paper presented here makes some excellent points, and should be encouraged.

The whole question of reproducibility was discussed some years ago, in areas from antibody validation, to biomarkers, and even protein coronas. The area is just so complex, I will summarise briefly.

I have some comments the authors might like to consider.

We thank the reviewer for meticulous reading of our paper and appreciating efforts like this one in trying to standardize scientific measurements. We have read the reviewers’ constructive comments and applied them in the discussion and other appropriate sections of the revised manuscript.

Firstly, many of these types of blind round robins were attempted earlier on in teams from across US and EU, and much was learned. Some of the conclusions were not very palatable, and as the work was sometimes not regarded as 'research' the reports were not always published in the most accessible places. In the nano area one of the most simple measurements and known round robins is Journal of Nanoparticle Research volume 15, Article number: 2101 (2013), and it is instructive to see how that was prepared, developed and done. Similar things were done for coronas, and many other biological outcomes such as toxicity.

We thank the reviewer for introducing these papers. We have carefully read and incorporated insights from these papers and discussed them in relevant sections of the revised manuscript. We totally agree with the reviewer that there are many records available in the literature that try to address the reproducibility and reliability of the data and research results in various fields (e.g., nanoscience and nanomedicine) like the article mentioned by the reviewer where authors discuss the size measurement variation across different laboratories and try to address the limitations and propose a standard protocol for size measurement of NPs (reference 37). A similar report (reference 40) discusses some examples of irreproducible reactions in chemistry and suggests the possible ways to increase such reproducibility. A nature nanotechnology article in 2019, focused on establishing guidelines and practices to facilitate translation of NPs by standardizing characterization methods, enhancing reproducibility between studies (reference 5). Our group also very recently reported on importance of standardizing analytical characterization methodology for improved reliability of the nanomedicine literature where we discussed the available standards for robust characterization of nanomaterials based on nanomedicine applications as well as the challenges ahead of such standard protocols and clinical translation of nanomaterials (reference 59). More discussions and citations in this regard have now been added to the manuscript, in light of your comments.

The outcomes were much as reported here, and a lot of efforts was put into understanding just why. Sadly, it transpired that even the most accomplished laboratories were often well off the standard answer in early phases, and this lead to a process of protocol and training between the engaged groups. This eventually lead to convergence between all of the labs for nearly every quantity-the particle size example illustrates, including high overlap for the protein corona lists of hits, and even rankings. This established that such outcomes were possible, but it was found that even the arduous efforts with protocols, in the absence of training between labs produced poor outcomes. It should be noted that the particle size example is one of the most simple, and many of the painters there invented the instruments, or developed them and were certainly outstanding! This leads to the uncomfortable that many measurements in science are poorly reproduced by teams of scientists, no matter how solid the lab, and that something extra is needed. This is a difficult message, and I think one hard for anyone to send, and likely it is implicit in what the authors are saying here. The most experienced labs in certain areas control an enormous number of factors within the labs, and others do not. Some labs get good results in certain biomarker assays, some do not until there is a process of protocol evolution, handover and training. All of this was also discussed at length within the Nature Reproducibility Initiative, and largely speaking all was clear.

We thank the reviewer for the valuable discussion. We have borrowed some of the insight on protocol evolution, handover and training and the importance of standardizing experimental work in our manuscript.

The question is how to pass the message in this particular example, and for this generation of people. Other report and many other scientists have shown how controls can be imposed (<https://hal.archives-ouvertes.fr/hal-02163525/document>, <https://www.ncbi.nlm.nih.gov/pmc/articles/PMC6631359/>). While I would not agree with all of the content of these many of these reports and discussions, the overall outcomes are sort of common and agreed. Certainly I think the authors do not wish to say that it is impossible to get reproducible answers between labs (we know this is not the case) nor that they as individuals are not able to succeed. After all, these are highly talented and excellent people, that could do so if they pushed ahead with the kind of rouble robin processes outlined above.

In response to reviewer response, we have further toned down our discussions on the reproducibility related statements throughout the revised manuscript. We also appreciate sharing of the papers, which we considered in revision of our work.

I am sure they have in mind that many results reported results are unreliable, and are illustrating this very well. Quite frankly, some labs are able to get good biomarker reproducibility in all fields and others are not, and the reasons are many. This is a very hard message to pass, as it can appear very critical. In my experience the truth of this issue is extremely difficult to communicate.

It will take maybe a little though to see how to refine the message of the paper to give some lasting value. I would allow the authors to refine their message a bit. But certainly they are making highly valuable points. And the fact the points have been made before and not fully absorbed may mean it is time for a good journal to take on the task to help?

We have taken note of the mentioned publications and incorporated a refined message in the current manuscript. As the reviewer has noted, we have tried to be very mild in our interpretation of results and have tried to restrict our interpretation within the protein corona field. As fully discussed in the manuscript, we have tried to find ways of encouragement for example by showing a good correlation for the shared proteins among the centers. We hope that the message of the paper will be received well and that the paper will also have a generally positive effect in the proteomics community, stimulating further discussions.

REVIEWERS' COMMENTS

Reviewer #1 (Remarks to the Author):

I suggest the publication of the manuscript in this revised form. The authors revised thoroughly the discussion providing a message which encourage a critical evaluation of the data obtained across different labs (which is of course an important point), as well as good practice in proteomics studies. This work will have certainly an impact in protein corona research.

Reviewer #2 (Remarks to the Author):

All my comments have been addressed by the authors in full. Therefore I feel that the paper is ready for publication as it is in its revised version.